# An interpretable automated detection system for FISH-based HER2 oncogene amplification testing in histo-pathological routine images of breast and gastric cancer diagnostics

**Sarah Schmell**[1,2]                                    SARAH.SCHMELL@ASGEN.DE
[1] *Institute of Pathology, University Hospital Carl Gustav Carus (UKD), TU Dresden, Dresden, Germany*
[2] *ASGEN GmbH & Co. KG, Dresden, Germany*

**Falk Zakrzewski**[1,2,3]                          FALK.ZAKRZEWSKI@UNIKLINIKUM-DRESDEN.DE
[3] *National Center for Tumor Diseases (NCT), Partner Site Dresden, Germany*

**Walter de Back**[4,5]
[4] *ContextVision AB, Stockholm, Sweden*
[5] *Institute for Medical Informatics and Biometry (IMB), Carl Gustav Carus Faculty of Medicine, TU Dresden, Dresden, Germany*

**Martin Weigert**[6,7,8]
[6] *Institute of Bioengineering, School of Life Sciences, EPFL, Lausanne, Switzerland*
[7] *Max Planck Institute of Molecular Cell Biology and Genetics (MPI-CBG), Dresden, Germany*
[8] *Center for Systems Biology Dresden (CSBD), Dresden, Germany*

**Uwe Schmidt**[7,8]
**Torsten Wenke**[2]
**Silke Zeugner**[1]
**Robert Mantey**[3]
**Christian Sperling**[1]
**Ingo Roeder**[3,4]
**Pia Hoenscheid**[1,3]
**Daniela Aust**[1,3]
**Gustavo Baretton**[1,3]

## Abstract

Histo-pathological diagnostics are an inherent part of the everyday work but are particularly laborious and associated with time-consuming manual analysis of image data. In order to cope with the increasing diagnostic case numbers due to the current growth and demographic change of the global population and the progress in personalized medicine, pathologists ask for assistance. Profiting from digital pathology and the use of artificial intelligence, individual solutions can be offered (e.g. detect labeled cancer tissue sections). The testing of the human epidermal growth factor receptor 2 (HER2) oncogene amplification status via fluorescence *in situ* hybridization (FISH) is recommended for breast and gastric cancer diagnostics and is regularly performed at clinics. Here, we develop an interpretable, deep learning (DL)-based pipeline which automates the evaluation of FISH images with respect to HER2 gene amplification testing. It mimics the pathological assessment and relies on the detection and localization of interphase nuclei based on instance segmentation networks. Furthermore, it localizes and classifies fluorescence signals within

each nucleus with the help of image classification and object detection convolutional neural networks (CNNs). Finally, the pipeline classifies the whole image regarding its HER2 amplification status. The visualization of pixels on which the networks' decision occurs, complements an essential part to enable interpretability by pathologists.

**Keywords:** FISH imaging, HER2 amplification, gastric/breast cancer, digital pathology, deep learning, image classification, object segmentation and localization, interpretability.

## 1. Introduction

The human epidermal growth factor receptor 2 (HER2) amplification status is an important tumor marker in breast and gastric cancer. It indicates a more aggressive disease with a greater rate of recurrence and mortality. Hence, it influences the decision making for finding an appropriate therapy (Mitri et al., 2012). The amplification of HER2 is detected by assessing fluorescence *in situ* hybridization (FISH) images: Fluorescence signals are counted in at least 20 interphase nuclei from tumor regions and are then graded into a HER2 negative or positive status with a high or low amplification (Wolff et al., 2018).

To optimize the diagnostics in terms of speed, accuracy, objectivity and interpretability, we are developing a comprehensible, multi-step deep learning (DL)-based pipeline (Figure 1A). It mimics the pathologist's evaluation steps and integrates the decision processes into a report for transparency (Figure 1A.6). The pipeline independently evaluates each nucleus twice by different networks creating a second opinion. The pre-selection of nuclei reduces the risks of isolating overlapping nuclei parts (e.g. signals) or artifacts which could alter the nucleus-specific classification. The first component is an instance segmentation network designed for cell containing data sets, called StarDist (Schmidt et al., 2018), to detect and extract all individual nuclei (not only 20) from the entire FISH slide (Figure 1A.1). The retrieved nuclei are then classified by a custom image classification convolutional neural network (CNN) and a RetinaNet-based FISH signal detector system (Lin et al., 2017) (Figure 1A.2 and 4). Visualizations, such as class activation maps (CAMs) (Zhou et al., 2016), display the decision making of the pipeline.

## 2. Methods

Selected tumor samples for our data sets originated from breast cancer tissues preserved in formalin-fixed paraffin-embedded (FFPE) blocks from clinical institutions all over Germany and were harbored at the Institute of Pathology of the Carl Gustav Carus Hospital of TU Dresden. FISH slides were produced and digitized as described in Zakrzewski et al. (2019). These routinely processed slides often display diverse artifacts, low signal-to-noise ratios and color disparities as a consequence of sample fixation and imaging.

All pipeline components were implemented in Python 3 using the Keras framework (version 2.2.4, Chollet et al. (2015)) and TensorFlow (version 1.12.0, Abadi et al. (2015)) as back-end. Trainings were performed on one Nvidia GeForce GTX 1080 Ti without pre-training.

The data set for the StarDist-based[1] (Schmidt et al., 2018) *Nucleus Detector* (Figure 1B, first panel) comprises randomly chosen images from 62 FISH slides from 2015 to

---

1. https://github.com/mpicbg-csbd/stardist

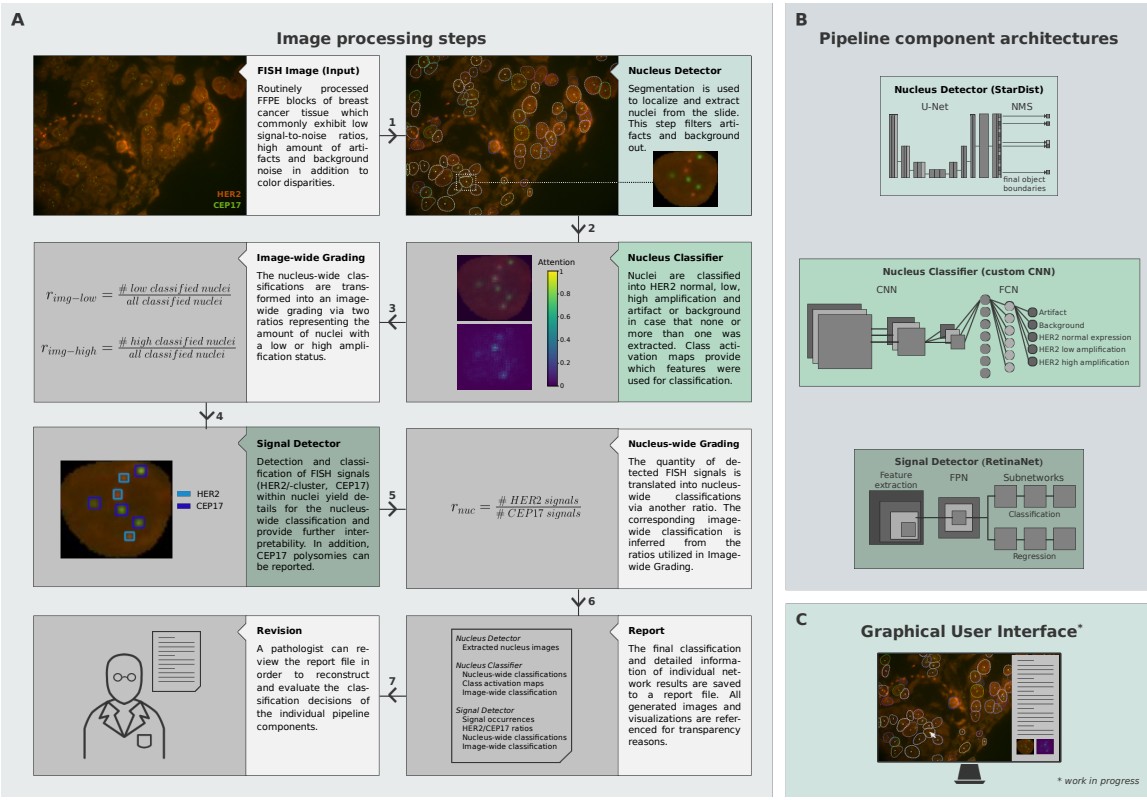

Figure 1: **(A)** Workflow for the automated evaluation of the HER2 gene amplification testing from FISH images. **(B)** Utilized architectures of the pipeline components. Adapted from Lin et al. (2017); Schmidt et al. (2018). **(C)** Graphical user interface for interactive access to the pipeline results for pathologists.

2018 (representing 62 patients, ∼7,440 nuclei, JPEG format) of the size 1,600 × 1,200 px. The nuclei were manually outlined by a pathologist using Fiji (Schindelin et al., 2012) and were exported as label masks[2]. Training was performed for 250 epochs with 1,000 steps per epoch using standard configurations and a learning rate scheduler. A validation split of 0.15 was used and the images were scaled down to half their size to fit the nuclei dimensions to the receptive field of the architecture. Additionally, augmentation operations including axis-aligned rotations and horizontal as well as vertical flips were utilized to increase the available number of training images. The test set included randomly chosen images (1,600 × 1,200 px) from 10 FISH slides (∼810 nuclei).

The *Nucleus Classifier* is a VGG-like (Simonyan and Zisserman, 2015) customized CNN of five convolutional and five fully connected layers interleaved with max pooling and batch normalization (Figure 1B, second panel). Adam optimizer (Kingma and Ba, 2015) and weighted categorical cross entropy were used. The data set consists of 8,313 single nucleus

---

2. https://forum.image.sc/t/creating-labeled-image-from-rois-in-the-roi-manager/4256/2

images (image dimensions varied around $80 \times 80$ px) randomly extracted from the *Nucleus Detector's* data set. These nucleus images were manually classified by a pathologist into artifact (277 images), background (6,439 images), HER2 normal expression (HER2 negative, 224 images) and HER2 low (HER2 positive, 362 images) or high amplification (HER2 positive, 224 images). The CNN was trained with a batch size of 16 due to GPU memory limitations for 300 epochs. The validation split was 0.2 (39 images, ∼316 signals). Images were not scaled down but augmented using the same operations as mentioned above.

The RetinaNet-based[3] (Lin et al., 2017) *Signal Detector's* (Figure 1B, third panel) data set is a subset of 397 single nucleus images (∼5,955 signals) of the *Nucleus Classifier's* set. The FISH signals within the nuclei were manually annotated by a pathologist into three classes: HER2 (single signal), HER2-Cluster (indeterminate amount of signals) or chromosome enumeration probe 17 (CEP17; reference centromeric satellite DNA, single signal). The training was performed with a ResNet50 (He et al., 2016) backbone and a validation split of 0.1 for 130 epochs using standard configurations. The images were not scaled down but augmented with brightness and contrast changes in addition to the operations above.

Network performances were estimated on validation and test sets using precision, recall and average precision (AP) scores.

## 3. Results

The first component, called *Nucleus Detector* (Figure 1A.1), was trained to detect interphase nuclei in the FISH image. The shape prediction of nuclei as star-convex polygons excludes classification-altering artifacts and overlapping nuclei parts. It achieves a precision score of 0.76 and recall score of 0.65 on the test set since the differentiation of small and adjacent nuclei (often predicted as single nucleus) remains challenging.

The *Nucleus Classifier* (Figure 1A.2) was used to classify the extracted nuclei into artifact, background, HER2 normal expression and HER2 low or high amplification. Thereby, the filter classes (artifact, background) ensure that images with no or more than one nucleus will not be considered for the HER2 amplification testing. On the validation set, the network achieves a precision and recall score of 0.98. CAMs (Figure 1A.2) are used to elucidate the classifications and demonstrate that the classes were recognized based on FISH signal presence and number.

The third component is the *Signal Detector* (Figure 1A.4) and was trained to localize and classify FISH signals within a single nucleus into HER2, HER2-cluster and CEP17. Thus, each nucleus is classified a second time (second opinion). The bounding boxes provide details regarding the number and position of FISH signals per nucleus. The *Signal Detector* achieves a mean AP of 0.73 on the validation set. The CEP17 signals (AP: 0.94) are detected very well but the detection and distinction of HER2 (AP: 0.65) and HER2-cluster signals (AP: 0.60) remains challenging. However, mainly crowded HER2 and weak FISH signals are not identified or detected in multiple and distinctly classified boxes.

The nucleus- and FISH image-wide HER2 amplification status is inferred via different ratios (Figure 1A.3 and 5) and thresholds as mentioned in Zakrzewski et al. (2019). The classification thresholds can be modified to the needs of any clinical purposes.

---

3. https://github.com/fizyr/keras-retinanet

All steps of the pipeline are documented in a report file, which can be used to evaluate the classifications made by the individual pipeline components (Figure 1A.6 and 7).

Our DL-based system is advanced and more comprehensible compared to the previous version of Zakrzewski et al. (2019) as it mimics the evaluation process of pathologists more closely including interpretability steps. Moreover, it is also capable of processing low quality FISH images characterized by high background noise, a large number of artifacts, low signal-to-noise ratio, weak signals, major differences in nuclei shape or overlapping nuclei often occurring in daily routine.

## 4. Conclusion

Our proposed pipeline is a step towards the development of a DL-based assisting tool for pathologists to cope with the growing number of cancer cases during clinical routine. It can be individually (re-)trained on lab-specific images to reach optimal performance. While increasing the speed of the evaluation, the pipeline additionally enhances objectivity, provides the maximum amount of information for medical reports and can be applied to any FISH-based (e.g. BCR/ABL, BCL/IGH fusions; MYC, BCL6, ALK translocations) analyses. To integrate these novel analysis requirements, it will be necessary to train our pipeline on the individual FISH protocol-specific images.

Potential implementation into clinical routines will be achieved using an intuitive interface (Figure 1C, web based, currently work in progress) to enable usage on a variety of devices (e.g. computer and tablet). The interface needs to be as easy to use as possible so that a minimum of extra training for pathologists is required. We are also optimizing our system to be applicable on whole slide images up to a size of $100\text{k} \times 100\text{k}$ px or more.

The source code for our pipeline can be found on GitLab[4].

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
