# OpenReview forum: "An interpretable automated detection system for FISH-based HER2 oncogene amplification testing in histo-pathological routine images of breast and gastric cancer diagnostics"
_MIDL.io/2020/Conference — MIDL 2020_

### Official Review · AnonReviewer2 · 2020-02-22
**An algorithmic pipeline for FISH based HER oncogene detection and quantification**

**Rating:** 3
**Confidence:** 4

**Review:**

Authors present a machine learning, computer vision pipeline for FISH based HER oncogene detection and quantification in histopath images.

The paper is well written and easy to follow.
Even if the technical contribution is limited, the main pitch of the paper is application novelty. The authors clearly present this in the paper and do not overclaim technical novelty.

The authors only provided performance for individual steps of the pipeline. End-to-end performance analysis should have been included.

The authors could have included a few more detail about the algorithm, such as the input size to individual networks, training parameters, etc..

---

### Official Review · AnonReviewer1 · 2020-03-05
**An interesting end-to-end pipeline for the interpretation of histo-pathological images**

**Rating:** 3
**Confidence:** 2

**Review:**

The authors describe a new end to end image analysis pipeline they developed to interpret histo-pathological images of breast and gastric cancers. Specifically, a deep convolutional neural network (CNN) first makes automatic the analysis of fluorescence in situ hybridization (FISH) images that test the Human Epidermal growth factor Receptor 2 (HER2) oncogene amplification status.  The deep learning pipeline mimics the pathological assessment, and localizes plus classifies the fluorescence signals within each nucleus. Then, it classifies the whole image regarding its HER2 amplification status.
This short paper gives a good overview of the pipeline and reads well. The methodology seems to be solid and very flexible. The results are also promising although the proposed pipeline is not compared with another state-of-the-art method. The source code of the pipeline is finally freely available. It therefore believe that this work will be of significant interest at MIDL.

---

### Official Review · AnonReviewer3 · 2020-03-09
**Hypothesis not answered, methods are unclear**

**Rating:** 1
**Confidence:** 5

**Review:**

The clinical problem this paper takes on is interesting and relevant for patient care, but the motivation is unclear. Is there a problem with the current method for evaluating HER2 status that deep learning can solve? The methods of this paper are unclear and it would be impossible to replicate this study from this manuscript. The hypothesis of this paper appears to be that deep learning can determine HER2 status, but it is unclear whether this was supported or refuted by these results.

Major comments:

The number of patients and number of images from each patient must be stated.

The paper refers to training, test, and validation sets, but the number of patients in each set and method by which they were allocated are unclear. Were the sets the same for each task?

Are all the patients from the same hospital?

The results section consists mostly of methods. There is no methods section.

It is unclear how artifacts or overlapping nuclear parts were excluded. It is unclear how this exclusion affected the results. If this exclusion is manual, it calls into question the claim of a fully automated pipeline.

How was the ground truth for these images established?

The performance metrics are given for each network individually, but it is unclear what the performance of the pipeline is on the overall task of patient HER2 classification. How does this performance compare to the current gold standard?

A main claim of this paper is that the pipeline is interpretable. However there is no description of the features used by any of the networks nor any biological insight provided by the networks. An interpretable network allows scrutiny of its classification decisions. It is unclear whether that is possible here.

Minor comments:

The test in figure 1 is so small that it becomes readable only at 200% size. It seems odd to have so much text in this figure rather than describing the process in the manuscript text and referencing the figure in the text.

The magnification and microns-per-pixel of the images should be given, as should the hardware used for digitization.

The inclusion of the code via github is good.

---

### Official Review · AnonReviewer4 · 2020-03-12
**Very well written**

**Rating:** 3
**Confidence:** 5

**Review:**

In this manuscript authors proposed a deep learning (DL)-based pipeline to automate the pathological assessment of FISH images with respect to HER2 gene amplification testing. Their pipeline detects  nuclei and classifies fluorescence signals within each nucleus using CNNs.
Pros:
-The paper is very well written and planned.
-The pipeline design is adequate.
-The experiments are well designed.
-The motivation and future direction given clearly.

Cons:
-Using term "interpretable" results is not adequate, as the results and generated report are not really interpretable. It can be maybe "human readable".
-It would be nice to discuss more on how this can be implemented in clinical practices, how much training is needed for practitioners

---

### Meta-Review · Area_Chair1 · 2020-03-31
**MetaReview of Paper188 by AreaChair1**

**Rating:** 3

**Metareview:**

Most reviewers suggest acceptance of the paper, whereas one reviewer suggests a strong rejection. The reviewers suggesting acceptance indicate that the paper is well-written, easy to follow and that the results look very promising. The pipeline is generally well-described and the tasks of the individual components is clear. I do agree with the reviewer recommending rejection that some important details are missing and that these could have been added to the paper (e.g. dataset splits). However, I think rejection would be too harsh, given that this is a short paper and quite and extensive method, the authors had to choose what to include and what not. As such, I lean towards acceptance.

**Paper Type:**

methodological development

---

### Decision · Program_Chairs · 2020-04-11

Accept